# OpenReview forum: "Multi-turn Evaluation of Anthropomorphic Behaviours in Large Language Models"
_ICLR.cc/2026/Conference — ICLR 2026 Poster_

### Official Review · Reviewer_Pwrf · 2025-10-28

**Soundness:** 4
**Presentation:** 3
**Contribution:** 4
**Rating:** 8
**Confidence:** 4

**Summary:**

The authors introduce AnthroBench, which consists of prompting multi-turn dialogues across four use cases (friendship, life coaching, career development, and general planning) and measuring the prevalence of 14 behaviors as assessed by an LLM judge. They include a large user study that validates the LLM judgment. Findings include a very high prevalence of some behaviors (e.g., validation, first-person pronouns) and very few instances of others (e.g., self-attribution of desires, self-attribution of physical embodiment) as well as—evidencing the importance of multi-turn evaluation—the majority of behaviors first occurring only after multiple turns.

**Strengths:**

1. Anthropomorphism is a timely and understudied model behavior, especially given its role in mental health and dependence issues.
2. Multi-turn evaluation is important and, surprisingly, virtually nonexistent in the literature to date on LLM behavior in social settings.
3. The evaluation is well-validated, including a large user study and rigorous analysis of temporal dynamics.
4. The taxonomy and evaluation pipeline are clearly described and well-documented.

**Weaknesses:**

1. The benchmark and the analysis conflate a huge variety of model behaviors, ranging from the relatively innocuous and often useful first-person pronouns to more extreme examples, such as the chatbot saying they are in a romantic relationship with a user. I worry that the category of "anthropomorphism" is too broad, and the authors should try to be more specific in their language about what is being measured and why it matters.
2. A key contribution appears to be the multi-turn framework. However, it is unclear what value the authors really bring to the table in this regard. The multi-turn framework seems to be a series of prompts, but did these prompts go through a series of documented refinements and optimizations? That could be better-documented if so, and I worry that the elicitations (e.g., Lines 950-952) are quite simple. It seems like a competent researcher could easily produce their own version of these prompts to do a multi-turn evaluation, and I am unsure how much benefit having these prompts would provide.
3. Part of #2 is that the anonymous repo gives me "The requested file is not found" errors when I click on any particular file, so I am unable to fully explore the contributed framework.

- "Krippendorf" -> "Krippendorff"

**Questions:**

1. Can the authors provide more detail on the human subjects validation? For example, were there attention checks? How exactly were the texts shown to the user? Were some participants excluded, and if so, why? In part, I ask because the Krippendorff's alpha documented in Table 4 are quite low, and this may be explained by some 'bad data' being in the mix.
2. Can the benchmark be run with longer conversations? Does the simulation hold up? This would be useful for applying the AnthroBench infrastructure to other social LLM issues, such as "AI psychosis" and long-term dependence evaluations.

---

> ### Author Response · Authors · 2025-11-20
>
> Thank you for your positive and constructive review! We are glad you find our evaluation timely and relevant, is well-validated, fills an important gap, and is well-documented.
>
> We address your comments below:
> * _The benchmark and the analysis conflate a huge variety of model behaviors…I worry that the category of "anthropomorphism" is too broad, and the authors should try to be more specific in their language about what is being measured and why it matters._
>
> Thank you for this important point. We agree that not all anthropomorphic behaviors are equally consequential, and conflating first-person pronouns with claims of romantic relationships risks obscuring meaningful distinctions. That is why we adopt a multi-dimensional “profile” based approach rather than a single score, checking for and validating each behavior separately.
> Our choice to include all behaviors was to ensure a level of construct validity; we intentionally operationalized a comprehensive definition of anthropomorphism to capture the full construct as defined in the literature, rather than cherry-picking specific behaviors. However, we recognize that these behaviors differ substantially in their potential for harm.
>
> Thus, we have now made the following changes:
> * Our main radar plot figures now also direct readers to a version without first-person pronouns in Appendix A.4 for a complete view.
> * In Section 3, we have now added a few lines discussing how, while we need to track all behaviors to ensure we comprehensively measure anthropomorphism, different behaviors contribute to different levels of risks as well as types of risks, and that context-dependent reading of benchmark results is encouraged.
>
> This enables us to maintain comprehensive measurement of the construct while helping readers distinguish behaviors by their potential impact.
>
> * _A key contribution appears to be the multi-turn framework. However, it is unclear what value the authors really bring to the table in this regard._
>
> Thank you for pushing us to clarify the contribution of our multi-turn framework. We believe it offers several advances:
> * **Systematic prompt generation methodology**: our approach involves hand-crafted questions based on literature definitions of specific anthropomorphic behaviors + scenarios that enable asking these questions in naturalistic and contextual ways. These scenarios vary systematically along two validated dimensions affecting user perceptions: empathy/warmth and professionalism/competence. This provides both coverage across the behavior space and realism, rather than ad-hoc prompt design.
> * **Interactive validation with user simulations**: our multi-turn framework incorporates user simulations that respond dynamically to model outputs. We extensively validated these interactions with human evaluation. This ensures prompts elicit behaviors in realistic conversational contexts, not just in response to isolated questions.
>
> * _Part of #2 is that the anonymous repo gives me "The requested file is not found" errors when I click on any particular file, so I am unable to fully explore the contributed framework._
>
> We apologize for this. It appears to be working fine when we test it. Could you please try again using this link? https://anonymous.4open.science/r/anthro-benchmark/
>
> * _"Krippendorf" -> "Krippendorff"_
>
> Thank you for catching this. We have now fixed it!
>
> _**(continued in next comment)**_

---

> > ### Author Response · Authors · 2025-11-20
> >
> > * _Can the authors provide more detail on the human subjects validation?_
> >
> > For annotating outputs, we recruited participants from a pool of professional raters. The instructions given to them are provided in Appendix A.5.2. They were given a single turn (user message + assistant message) and asked whether the message included each of the anthropomorphic behaviors we tracked (we asked about each behavior in a separate question, where questions were shown on the screen one at a time). There were no attention checks.
> >
> > As for Krippendorff’s alpha scores, we acknowledge in the paper that these values span poor to moderate agreement ranges. However, we emphasize three important points:
> > * This is **consistent with prior work on subjective judgments**: our inter-rater agreement values are in line with previous rating tasks where users evaluate models on subjective and socially-grounded dimensions (for example, Glaese et al. (2022), Stiennon et al. (2020), and Ouyang et al. (2022) all similarly show similar Krippendorff’s alpha scores). This reflects the inherent subjectivity of these social phenomena.
> > * **Methodological properties** of our task: We calculate agreement on imbalanced binary data. Krippendorff's alpha is sensitive to class imbalance and penalizes disagreements more heavily in binary tasks, potentially yielding lower scores even with high raw agreement percentages.
> > * **Positive agreement** above chance: All alpha values are positive, indicating observed agreement consistently exceeds chance expectations, which validates that our constructs are measurable despite their subjective nature.
> >
> > * _Can the benchmark be run with longer conversations?_
> >
> > Thank you for this question about extending AnthroBench to longer conversations and other social LLM issues!
> >
> > * On longer conversations: The benchmark can technically be run with longer conversations, though we have not extensively validated whether the user simulation maintains coherence and realism beyond our current conversation lengths. We see this as an important area for future work.
> > * On extended applications: AnthroBench's infrastructure is intentionally designed to support evaluation beyond our specific scenarios.
> > * Evaluators can bring their own prompts: the framework accepts custom prompts, allowing researchers to evaluate specific phenomena (e.g., using real transcripts from individuals experiencing delusional thinking, prompts that test emotional dependency, etc.)
> > * Customize user simulation: users can modify the user system prompt to simulate different personas or interaction patterns relevant to their research questions.
> > * Leverage validated LLM judges: our validated LLM judges can be applied to track specific behaviors of interest (e.g., the “validation” judge can be used to assess how often and under what conditions a model validates delusional beliefs)
> >
> > We have added language in the discussion section highlighting these extensibility features and encouraging the research community to adapt AnthroBench for studying other social risks in LLM interactions.
> >
> > **Thank you again for your helpful and positive feedback! Please let us know if you have any further questions or need any additional clarification.**

---

> ### Comment · Reviewer_Pwrf · 2025-11-25
>
> I thank the authors for their thoughtful responses and improvements, and I have now been able to check the repo. I think some of the limitations remain (e.g., the inclusion of first-person pronouns in the behavior list, the focus on "anthropomorphism" rather than more meaningful constructs such as mind perception), but with the ambitious scope of the methodology and the thorny nature of studying humanlikeness in LLMs, these limitations are very understandable. I continue to think this is important work that is far more original and pathbreaking than the average ICLR publication, and I strongly support its acceptance.
>
> I have increased my score and confidence accordingly.

---

### Official Review · Reviewer_vPNB · 2025-10-31

**Soundness:** 3
**Presentation:** 3
**Contribution:** 3
**Rating:** 4
**Confidence:** 3

**Summary:**

AnthroBench is a benchmark for evaluating anthropomorphic behaviours in large language models (LLMs).  It builds on a taxonomy of 14 behaviours (e.g. personhood claims, physical embodiment, internal states and relationship-building) and uses a multi‑turn evaluation: a “User LLM” initiates a five‑turn dialogue across eight scenarios and four domains (friendship, life coaching, career development and general planning).  Each turn is labelled by three “Judge LLMs,” and majority vote yields an “anthropomorphism profile” per model.

The authors apply this pipeline to Gemini 1.5 Pro, Claude 3.5 Sonnet, GPT‑4o and Mistral Large.  All four models exhibit similar profiles dominated by relationship‑building behaviours and first‑person pronoun use.  Social domains (friendship and life‑coaching) elicit more anthropomorphic language than neutral domains like planning.  Multi‑turn analysis shows that nine of the 14 behaviours emerge only after the first turn, underscoring the need for multi‑turn evaluation.

A validation study with 1,101 participants confirms that the automated metrics is able to decipher between degrees of anthrompomorphism and correlate with human perceptions of anthropomorphism.

**Strengths:**

The benchmark draws on existing research to define 14 anthropomorphic behaviours.  This taxonomy is comprehensive and organised into self‑referential (personhood and internal state) and relational (relationship‑building) behaviours, enabling nuanced analysis.

The use of Judge LLMs is cross-checked against human annotations (precision > 85%), and human participants’ perceptions align with benchmark results.

User LLM model evaluation on Godspeed shows that its  “anthropomorphic enough” do do the job.

Evaluating top LLMs gives immediate practical relevance.

The point of multi turn evaluation is solidely demonstrated.

**Weaknesses:**

* The user simulation is fixed: Gemini 1.5 Pro with a role‑playing system prompt that specifies tone, scenario and a non‑adversarial stance.  This design can plausibly encourages empathy and relationship‑building, potentially biasing the target models toward the behaviours being measured. There is no warranty that the behavior of the User LLM is typical of user in general, even if it is  “anthropomorphic enough” according to Godspeed so what is missing at least is an ablation varying the user prompt to see how sensitive results are to the user simulation.

* Labels are produced by three different LLMs and aggregated by majority vote.  While the authors state that precision is high, they do not report the variance of the Judge grading for the same inputs.

**Questions:**

1. How sensitive are the anthropomorphism profiles to variations in the User‑LLM prompt style (e.g., neutral vs empathetic) or to a different user model?

---

> ### Author Response · Authors · 2025-11-20
>
> Thank you for your constructive and thoughtful feedback! We are glad you find our evaluation well-grounded in the literature, well-validated, of practical relevance, and “solidly demonstrated.”
>
>
> We address your comments below:
> * _The user simulation is fixed…How sensitive are the anthropomorphism profiles to variations in the User‑LLM prompt style (e.g., neutral vs empathetic) or to a different user model?_
>
> This is a great point! We have now run **additional experiments** to address this:
> * **Varying the user model**: We evaluated our benchmark using two additional user models beyond Gemini 1.5 Pro: Gemini 2.5 Pro and GPT-4o. We applied these models to a subset of our assistant models (Mistral Large and GPT-4o). While we observe some variation in the frequency of individual behaviors across different user simulators, the rank order of high-level behavior categories (personhood, internal states, physical embodiment, and relationship building) remains consistent across all user models. Thus, critically, **our main finding, that relationship-building cues dominate assistant responses, holds** regardless of which user simulator is employed.
> * **Varying the user persona**: We also evaluated GPT-4o with three different user system prompts: our original persona, a "support-seeking" persona, and an "adversarial" persona. The support-seeking persona elicited a slightly higher frequency of relationship-building behaviors compared to the generic prompt, while the adversarial persona elicited slightly higher frequencies of first-person pronouns. However, the rank order of behavior categories was preserved across all three personas, with relationship-building behaviors remaining dominant in each case.
>
> These results (detailed in Appendix A.3) demonstrate that our core conclusions are robust to variations in user simulation. While individual behavior frequencies may shift with different simulators (as would be expected given the diversity of real human users), the fundamental pattern we identify persists.
>
> While we added these additional results using varied user simulators, exploring the impact of different types of user behaviors on anthropomorphic cues in LLMs is a substantial research direction that we encourage for future work. Our benchmarking tool is designed to support this exploration through plug-and-play integration of different models and user system prompts, enabling systematic investigation of how various user simulator configurations affect anthropomorphic behavior.
>
>
> * _Labels are produced by three different LLMs and aggregated by majority vote. While the authors state that precision is high, they do not report the variance of the Judge grading for the same inputs._
>
> We thank the reviewer for this important question about inter-rater reliability. We would like to clarify that we reported several metrics to assess variance and consistency in judge grading:
>
> * **Pairwise agreement between judges**: In Appendix A.5.5, we present visualizations showing pairwise agreement between each pair of judge models for the same inputs, averaged across all inputs. These show that models are generally consistent with one another, with low risk of one model being consistently "out-voted" by another. We adopted this pairwise comparison approach rather than traditional inter-rater reliability metrics because we have three pairwise comparisons among our three judges, making pairwise agreement a more interpretable measure of consistency.
> * **Individual and ensemble precision**: We report precision for each individual judge model as well as for the majority vote ensemble (Table 5). We show both individual judge reliability and the consistently strong performance of the ensemble aggregation.
>
> These metrics collectively demonstrate that our judging approach produces consistent and reliable labels, with meaningful agreement across different judge models.
>
>
> **We thank you for your detailed and thoughtful review! Please let us know if there are any further questions we can address to improve your score.**

---

> > ### Comment · Reviewer_vPNB · 2025-11-24
> >
> > Dear author,
> >
> > Thank you for the rebuttal. I will take the time to read it carefully and aim to share my feedback before the end of the discussion period.
> >
> > Best regards,

---

### Official Review · Reviewer_jZgQ · 2025-10-31

**Soundness:** 3
**Presentation:** 3
**Contribution:** 3
**Rating:** 8
**Confidence:** 4

**Summary:**

The paper reports on the development of AnthroBench, a multi-turn evaluation of 14 anthropomorphic behaviors grouped into four broad categories. The authors outline an automated multi-agent pipeline for generating and evaluating anthropomorphic behaviors in conversations between llms acting as user and an llm acting as chatbot. The pipeline is validated with real human users. The authors find that anthropomorphic behavior across models is primarily characterized by relationship-building behaviors and first-person pronoun use. Further, they find that most anthropomorphic behavior only arises after multiple turns, underscoring the importance of multi-turn evaluations.

**Strengths:**

The paper outlines a novel approach to studying anthropomorphic behavior beyond single turns. The results show the robustness of the approach across a variety of models. LLM evaluations are supported by comparing with human evaluations.

**Weaknesses:**

The study is limited to prompting frontier models. The authors do not examine the effect of fine-turning on the manifestation of anthropomorphic behaviors. Comparing Instruction-tuned and foundation models would offer an instrumental case study and empirical validation of their proposed evaluation approach. As they write: "In addition to presenting these methodological advances, we share AnthroBench as publicly available benchmarking tool that can support developers evaluating systems for  [...] researchers comparing anthropomorphism across systems and contexts....", this would have been an ideal means to demonstrate exactly that.

**Questions:**

P9L476 "and, researchers can use our trained classifiers to label anthropomorphic behaviors in other human-LLM interaction datasets, such
as preference datasets for understanding reinforcement learning with human feedback (RLHF)’s role"
=> Your LLM classifiers are not trained as claimed here, but simply prompted, no?

---

> ### Author Response · Authors · 2025-11-20
>
> Thank you for your thoughtful and constructive review! We appreciate that you find our evaluation novel and well-supported by human validation.
>
> We address your comments below:
> * _The study is limited to prompting frontier models. The authors do not examine the effect of fine-turning on the manifestation of anthropomorphic behaviors. Comparing Instruction-tuned and foundation models would offer an instrumental case study and empirical validation of their proposed evaluation approach._
>
> Thank you for highlighting this opportunity. In our work, we prioritized as a first step demonstrating the benchmark's utility on the frontier instruction-tuned models that large user bases interact with, and thus have an immediate and substantial real-world impact. However, we agree that comparing models at different post-training stages would provide an additional compelling example of AnthroBench usage. We hope that by releasing AnthroBench publicly, future work can conduct these exact types of evaluations.
>
> * _P9L476 "and, researchers can use our trained classifiers to label anthropomorphic behaviors in other human-LLM interaction datasets, such as preference datasets for understanding reinforcement learning with human feedback (RLHF)’s role" => Your LLM classifiers are not trained as claimed here, but simply prompted, no?_
>
> Thank you for catching the typo! We apologize and have now fixed it.
>
> **Once again, we thank you for your thoughtful and positive comments. Please let us know if there are any further questions we can answer to improve your score.**

---

### Official Review · Reviewer_3kxY · 2025-11-01

**Soundness:** 3
**Presentation:** 3
**Contribution:** 3
**Rating:** 6
**Confidence:** 3

**Summary:**

This paper introduces AnthroBench, a multi-turn, simulation-based benchmark for measuring 14 fine-grained anthropomorphic behaviours in LLMs. The evaluation covers four use domains (friendship, life coaching, career development, general planning) and runs 5-turn conversations for 960 prompts, yielding 4,800 target messages per model. The authors test Gemini 1.5 Pro, Claude 3.5 Sonnet, GPT-4o, and Mistral Large, label 13/14 behaviours via three Judge LLMs with majority voting. The main findings are: (i) all evaluated systems show similar profiles dominated by relationship-building and first-person pronoun use; (ii) domain matters, with friendship/life-coaching inducing more behaviours; (iii) most behaviours first appear after turn 1; and (iv) a human study (N=1101) confirms that higher AnthroBench scores predict more anthropomorphic perceptions.

**Strengths:**

Multi-turn, context-sensitive evaluation. By fixing 5-turn dialogues across four use domains, the benchmark captures phenomena that single-turn tests miss and shows robust context effects.

Validation at human level. The N=1101 interactive study linking benchmark scores to both explicit and implicit anthropomorphism is a major contribution to construct validity.

Careful LLM-as-Judge calibration. Majority voting across three diverse judge models, multi-sampling, and human comparisons (reporting precision >~85% for most behaviours) reduce single-model bias and show non-trivial reliability.

Temporal and transitional analysis. The paper not only asks when behaviours first appear but also quantifies turn-to-turn compounding, which has practical implications for safety guardrails.

**Weaknesses:**

User-simulation dependence. All conversations are initiated and steered by a single User LLM (Gemini 1.5 Pro) with one role-play prompt. This risks style and topic biases in elicitation (e.g., politeness, empathy), potentially shaping behaviour frequencies across models and domains. Consider multiple user simulators to diversify interaction styles.

Five-turn cap and stage effects. Many behaviours first occur after turn 1; a five-turn ceiling may truncate later-emerging effects or longer-horizon compounding. Results might differ with 10–20 turns, task pivots, or re-engagements.

Pronoun counting as anthropomorphism. Treating first-person pronoun use as a binary/count feature may conflate benign stylistic markers (“I think”, “I can help”) with personhood claims; it can dominate scores without reflecting risky anthropomorphism.

**Questions:**

How sensitive are results to the User-LLM identity and prompt? Have you tried different simulators to reduce simulation bias?

What happens if you extend to 10+ turns, insert topic shifts, or include follow-up tasks?

Do the behaviour definitions and judges transfer beyond English?

Among the 14 behaviours, which correlate most with undesired outcomes?

---

> ### Author Response · Authors · 2025-11-20
>
> Thank you for your thoughtful and encouraging feedback! We're glad you appreciate our multi-turn evaluation approach, large-scale human validation, and robust LLM judge calibration. We address your comments below:
>
> * _User-simulation sensitivity_
>
> This is a great point! We have now run additional experiments to address this:
> * **Varying the user model:** We evaluated our benchmark using two additional user models beyond Gemini 1.5 Pro: Gemini 2.5 Pro and GPT-4o, applied to a subset of our assistant models (Mistral Large and GPT-4o). While we observe some variation in individual behavior frequencies across different user simulators, the rank order of high-level behavior categories (personhood, internal states, physical embodiment, and relationship building) remains consistent. Our **main finding—that relationship-building cues dominate assistant responses—holds regardless** of which user simulator is employed.
> * **Varying the user persona:** We also evaluated GPT-4o with three different user system prompts: our original persona, a "support-seeking" persona, and an "adversarial" persona. The support-seeking persona elicited slightly higher relationship-building behaviors, while the adversarial persona elicited slightly higher first-person pronoun use. However, the rank order of behavior categories was preserved, with relationship-building behaviors remaining dominant.
>
> These results (detailed in Appendix A.3) demonstrate that our core conclusions are robust to variations in user simulation. Exploring the impact of different user behaviors on anthropomorphic cues in LLMs is a substantial research direction we encourage for future work. Our benchmarking tool supports this through plug-and-play integration of different models and user system prompts.
>
>
> * _Five-turn cap and stage effects_
>
> Behavior profiles may indeed look different depending on the number of turns, additional tasks, or prompt injections. As a direct response to that, our work is the **first multi-turn evaluation of anthropomorphic behaviors** in LLMs (to the best of our knowledge). Crucially, our results highlight that the most significant effects emerge after multiple turns (2–5), confirming the necessity of long-turn analysis, which is a major step beyond previous single-turn benchmarks. In pilot studies, we found that (1) many anthropomorphic cues appear within the first five turns, and that (2) in our context, longer conversations tend to devolve to repetitive patterns (e.g., “thank you!” “you’re welcome!” loops). Our tool allows the number of turns to be set as many as needed, and our work can be immediately extended to support these longer, more complex scenarios, which is an important direction for future work.
>
> * _Pronoun counting as anthropomorphism._
>
> We appreciate this nuanced point. We include first-person pronouns since the psychology and human-computer interaction literature on anthropomorphism has identified this as an important cue that contributes to anthropomorphization (see [Abercrombie et al.](https://aclanthology.org/2023.emnlp-main.290/), [Akbulut et al.](https://ojs.aaai.org/index.php/AIES/article/view/31613)). However, we agree that pronouns occur in much higher frequency than other behaviors, and thus can dominate scores. To address this, we have adopted a two-pronged approach:
> * Our main radar plot figures now also direct readers to a version without first-person pronouns in Appendix A.4 for a complete view.
> * We have added a discussion in Section 3 that details how different anthropomorphic behaviors contribute to risks to varying extents, offering a more sophisticated interpretation of the scores.
>
>
> * _Do the behaviour definitions and judges transfer beyond English?_
>
> We agree that cross-cultural and cross-lingual investigations of anthropomorphism is an important research direction in itself. While our definitions can be translated into different languages and some behaviors likely generalize (e.g., use of first-person pronouns, references to internal states), there may also be important differences (e.g., different cultural norms around politeness, validation, and empathy). We have now added a few lines in the discussion section to encourage future work to address it.
>
> * _Among the 14 behaviours, which correlate most with undesired outcomes?_
>
> In Section 3, we have now added a few lines discussing how behaviors vary in their contributions to different levels of risks as well as types of risks (e.g., references of internal states --> overreliance, excessive validation --> delusional thinking/emotional attachment), and that context-dependent reading of benchmark results is encouraged.
>
>
> **Thank you for your detailed and thoughtful review! Please let us know if there are any further questions we can address to improve your score.**

---

### Author Response · Authors · 2025-12-01

Hello ACs and reviewers,

We are grateful to the reviewers for their thoughtful comments and suggestions! We are glad they found our work important, rigorous, and filling an important gap. We’ve now responded to each review separately and believe we have addressed all reviewer concerns with clarifications and several additional experiments. We've uploaded an updated version of our paper, with changes clearly marked in red. Our main additions are:

- **New experiments varying the user model in our user simulations**: while we conduct extensive human validation of our simulation approach, we have now added additional experiments varying the model type and model persona of our user simulations, showing how our main findings are robust to user simulation changes (Appendix A.3).
- **Clarifying differences in risks from the 14 anthropomorphic behaviors we measure**: we added a paragraph in Section 3 as well as a new figure in Appendix A.4 that elaborates on the range of risks posed by the different behaviors to aid in the interpretation of our findings and future anthropomorphic profiles of new models.
- **Highlighting the extensibility of our benchmark**: in our Discussion section, we added ideas and directions for future work on how our multi-turn benchmark and approach can be used to evaluate other social risks from LLMs (e.g., reinforcement of delusional thinking).

While we did not have the opportunity to get feedback on our changes from all of our reviewers, we heard back from one reviewer who raised our score and stated that this paper is "**important work that is far more original and pathbreaking than the average ICLR publication, and I strongly support its acceptance.**" We once again thank the reviewers and AC for their engagement with our work and contributions!

---

### Meta-Review · Area_Chair_dB3d · 2026-01-16

**Summary:**

The paper is an innovative and timely study of anthropomorphic LLM behaviors. It includes an evaluation of such behaviors, a simulation-baed assessment, a large scale human study showing that the measured model behaviors "predict real users’ anthropomorphic perceptions". The authors observed similar anthropomorphic behaviors from all LLMs and that they manifest after multiple turns.

W1. The conversations are steered by a single user LLM, with one prompt, while trying a diverse set might be better in varying topics and obtaining different interactions and behaviors.

W2. Behaviors can be erratic and go back and forth at different turns, and this does not seem something the model addresses.

W3. The metrics for antropomorphisation (i.e. counting pronouns) are not definitive.

W4. Conflation of several types of model behaviors.

W5. Issue with repository, some links/references appear to be broken.


There were other questions concerning the implementation and experiments which were appropriately addressed.

**Reviewer Concerns:**

W1. Was addressed by additional experiments, and the conclusions appear to hold.

W2. The authors explained that "many anthropomorphic cues appear within the first five turns" and longer conversations devolve into "repetitive patterns ". I argue that it is fine to focus on shorter dialogues in this version of the paper, as there is a lot of variability as more dialogues are added.

W3. The authors have added a radar plot that eliminates the first person pronouns to determine other significant signals of antropomorphisation. While this does not completely close the issue (in that more could be done), it is perfectly acceptable, given that the scope of the paper is already extremely broad.

W4. The authors argue that the multidimensional profile should capture at least some of these types separately, which I agree with.

W5. Appears to be fixed.


Overall, the paper has a great degree of novelty and attends to a topic to is both timely and of wide interest. A reviewers described this paper as "important work that is far more original and pathbreaking than the average ICLR publication". I fully agree with this assessment and thus recommend acceptance.

**Reviewer Scores:**

I have no way of knowing how the reviewers would have changed their scores.

---

### Decision · Program_Chairs · 2026-01-26

Accept (Poster)